# Coaggregation Occurs between a Piliated Unicellular Cyanobacterium, *Thermosynechococcus*, and a Filamentous Bacterium, *Chloroflexus aggregans*

**DOI:** 10.3390/microorganisms12091904

**Published:** 2024-09-19

**Authors:** Megumi Kono, Shin Haruta

**Affiliations:** Department of Biological Sciences, Tokyo Metropolitan University, 1-1 Minami-Osawa, Hachioji 192-0397, Tokyo, Japan; kono-megumi@ed.tmu.ac.jp

**Keywords:** cell-to-cell interaction, cell aggregation, cyanobacteria, type IV pili, filamentous anoxygenic photosynthetic bacteria, gliding motility

## Abstract

Cyanobacteria are widely distributed in natural environments including geothermal areas. A unicellular cyanobacterium, *Thermosynechococcus*, in a deeply branching lineage, develops thick microbial mats with other bacteria, such as filamentous anoxygenic photosynthetic bacteria in the genus *Chloroflexus*, in slightly alkaline hot-spring water at ~55 °C. However, *Thermosynechococcus* strains do not form cell aggregates under axenic conditions, and the cells are dispersed well in the culture. In this study, *Thermosynechococcus* sp. NK55a and *Chloroflexus aggregans* NBF, isolated from Nakabusa Hot Springs (Nagano, Japan), were mixed in an inorganic medium and incubated at 50 °C under incandescent light. Small cell aggregates were detected after 4 h incubation, the size of cell aggregates increased, and densely packed cell aggregates (100–200 µm in diameter) developed. Scanning electron microscopy analysis of cell aggregates found that *C. aggregans* filaments were connected with *Thermosynechococcus* sp. cells via pili-like fibers. Co-cultivation of *C. aggregans* with a pili-less mutant of *Thermosynechococcus* sp. did not form tight cell aggregates. Cell aggregate formation was observed under illumination with 740 nm LED, which was utilized only by *C. aggregans*. These results suggested that *Chloroflexus* filaments gather together via gliding motility, and piliated cyanobacterial cells cross-link filamentous cells to form densely packed cell aggregates.

## 1. Introduction

Oxygenic photoautotrophic bacteria, cyanobacteria play an important role as a primary producer on Earth [1]. In natural environments, cyanobacteria form cell aggregates and biofilms [2]. Multicellular assemblage formation increases tolerance against environmental stresses, such as desiccation, osmotic pressure, and ultraviolet irradiation [3]. In cell aggregates and biofilms, cyanobacterial cells secrete extracellular polymeric substances, such as polysaccharides, proteins, and nucleic acids [2], and synthesize pili [4] to connect cells with each other. In cyanobacterial cell aggregates, coexistence of other bacteria is frequently observed [5]. The co-aggregate formation of cyanobacteria and other heterotrophic bacteria not only increases stress tolerance but also promotes their metabolic exchange (e.g., photosynthetic product and vitamins) [6,7]. However, cell-to-cell interaction between cyanobacteria and others in cell aggregates has not been studied well.

At geothermal springs, cyanobacteria-dominated biofilms are often observed under hot-spring water [8,9,10,11]. In slightly alkaline hot-spring water from 50 °C to 65 °C, a deeply branched cyanobacterial group, *Thermosynechococcus* (H. Katoh, S. Itoh, J.-R. Shen, and M. Ikeuchi [12]), is widely distributed as a dominant species worldwide [13,14]. *Thermosynechococcus* cyanobacteria have been studied in photosynthetic apparatuses [15,16,17,18,19,20,21] and their tactic behavior through type IV pili-mediated twitching motility [22]. Its CO_2_-fixing ability also attracts attention as a cell factory [23]. The metabolic network and material cycle in well-developed multispecies thick biofilms (~0.5 cm thickness, so-called microbial mats), dominated by *Thermosynechococcus*, are of great interest as a model ecosystem driven by oxygenic photosynthesis [24,25,26,27]. However, it has not been clarified yet how densely packed mats are formed and how the stratified structure of mats is developed. Unicellular *Thermosynechococcus* cells do not show adherability to solid surfaces and do not form cell aggregates under growing conditions, although cell aggregate formation is induced by environmental stresses, such as low temperature and blue light [28,29]. 

Castenholz suggested cell aggregate formation of unicellular, rod-shaped cyanobacteria with filamentous anoxygenic photosynthetic bacteria in the genus *Chloroflexus* in Hunter’s Hot Springs in Oregon, USA [11]. Bacteria in the genus *Chloroflexus* (Pierson and Castenholz [30]) have been characterized as a fast-gliding bacteria [31]. The representative species, *Chloroflexus aggregans* (Hanada et al. [32]), forms cell aggregates through active gliding motility on other filaments [33], but the aggregation is easily dispersed because there is no adhesive attachment between the filaments. *C. aggregans* was the second dominant species in the *Thermosynechococcus* microbial mats at Nakabusa Hot Springs (Nagano, Japan), one of several well-studied thermal springs (slightly alkaline, pH ~8.5; sulfidic, ~0.1 mM; up to 94 °C) [25,26]. It was hypothesized that *C. aggregans* helps form cell aggregates of *Thermosynechococcus* through cell-to-cell interactions. 

In this study, a piliated cyanobacterium, *Thermosynechococcus* sp. NK55a [34], and a filamentous bacterium, *C. aggregans* NBF [35], isolated from Nakabusa Hot Springs, were co-cultivated under gently flowing water conditions to investigate their cell aggregate-forming ability.

## 2. Materials and Methods

### 2.1. Bacterial Strains

*Thermosynechococcus* sp. strain NK55a (=NBRC 108920) [34], isolated from Nakabusa Hot Springs (Nagano, Japan), was cultured at 50 °C in a 60 mL BG11 medium [36] in 100 mL glass flasks with a cellulose cap (Steristopper, Heinz Herenz, Hamburg, Germany), with shaking at 80 rpm under illumination using a fluorescent lamp (15~20 μmol/s/m^2^). *C. aggregans* NBF, isolated from Nakabusa Hot Springs [35], was anaerobically cultured at 50 °C in a heterotrophic, 1/5PE medium [32,37,38] under illumination using an incandescent lamp (15~20 μmol/s/m^2^). Anaerobic conditions were achieved by completely filling a screw-capped glass test tube (∅18 mm, 32 mL volume). *Thermosynechococcus* and *Chloroflexus* growth was monitored by measuring the optical density at 740 nm with UV-1800 (Shimadzu, Kyoto, Japan) and 660 nm with miniphoto 518R (Taitec, Koshigaya, Japan), respectively. The number of *Thermosynechococcus* cells in the culture was assessed using a counting chamber (SLGC, Saitama, Japan) with a phase-contrast microscope (AXIO Imager A2; Carl Zeiss, Oberkochen, Germany). 

### 2.2. Disruption of the pilB Gene in Thermosynechococcus sp. NK55a

For homologous recombination, a plasmid, pUC19-∆pilB_CmR (8804 bp), harboring the upstream and downstream regions of the *pilB* gene (NK55_09820) in the NK55a genome [34] and chloramphenicol-resistant cassette, was constructed to transform strain NK55a. Based on the genomic information of *Thermosynechococcus* sp. NK55a (NC_023033), PCR primer sets (Table 1) were designed using a web tool (primer-BLAST; https://www.ncbi.nlm.nih.gov/tools/primer-blast/, accessed on 24 April 2023) to amplify the upstream (2.7 kbp) and downstream (2.6 kbp) regions of the *pilB* gene using NK55a genomic DNA as the template, which was extracted using a DNeasy UltraClean microbial kit (Qiagen, Venlo, The Netherlands). DNA fragments of the chloramphenicol-resistant cassette and a vector backbone were amplified from pHSG396 (Takara Bio, Kusatsu, Japan) and pUC19 (Takara Bio), respectively. PCR was performed using primerSTAR (Takara Bio) and primers listed in Table 1 according to the following protocol (32 cycles): denaturation, 95 °C for 10 s; annealing, 55 °C for 5 s; and polymerization, 72 °C for 10 s (chloramphenicol-resistant cassette) or 20 s (upstream and downstream regions of the *pilB* gene). DNA fragments were purified using the QIAEXII Gel Extraction kit (Qiagen) after agarose gel electrophoresis. Four fragments (vector backbone, upstream region of *pilB*, chloramphenicol-resistant gene cassette, and downstream region of *pilB*) were assembled using the HiFi DNA assembly master mix kit (New England Biolabs, Ipswich, MA, USA) according to the manufacturer’s instructions. Assembled DNA was used to transform *Escherichia coli* DH5α competent cells (Takara Bio) according to the manufacturer’s instructions. Construction of the obtained plasmids was confirmed by restriction enzyme digestion and DNA sequencing (BigDye Terminator kit v3.1 and SeqStudio Genetic Analyzers, Thermo Fisher Scientific, Waltham, MA, USA). The natural transformation of *Thermosynechococcus* sp. NK55a was performed according to Iwai et al. [39]. Mutant strains were selected by cultivation at 45 °C in BG11 agar medium containing 5 µg/mL chloramphenicol. Complete segregation via the chloramphenicol-resistant gene cassette was confirmed by DNA sequencing using the primers listed in Appendix A.

### 2.3. Transmission Electron Microscopy (TEM) Imaging

Bacterial cells were fixed with 2% glutaraldehyde in 0.1 M of cacodylate buffer (pH 7.4) for 1 min on carbon-coated Cu grids. After that, the samples were stained with 2% phosphor tungstic acid solution (pH 7.0) for 30 s and observed with a transmission electron microscope at 100 kV (JEM 1400Plus; JEOL, Akishima, Japan) at Tokai Electron Microscopy (Nagoya, Japan).

### 2.4. Co-Cultivation of Thermosynechococcus sp. and C. aggregans 

Cells at the late exponential growth phase (OD740 = 0.6–0.8 for *Thermosynechococcus* sp. and OD660 = 0.4–0.6 for *C. aggregans*) were harvested by centrifugation at 7600× *g* for 5 min and suspended together into a 5 mL BG11 medium (OD740 = 0.48 for *Thermosynechococcus* sp. and OD660 = 0.30 for *C. aggregans*). The cell mixture was added to a 25 mL glass vial, which was sealed with a butyl rubber stopper and aluminum cap. The vials were incubated at 50 °C in the light (incandescent lamp, 15–20 μmol/s/m^2^) using a roller apparatus at 20 rpm (VMRC-5; AS ONE, Osaka, Japan). When indicated, 630 and/or 740 nm LED arrays were used in place of incandescent lamps. 

### 2.5. Bright-Field and Fluorescence Microscopy Imaging 

Bright-field and fluorescence microscopic images of bacterial cultures were captured with a microscope (AXIO Imager A2; Carl Zeiss) equipped with a color digital camera, DP73 (Olympus, Tokyo, Japan), and CellSens standard 1.18 software (Olympus). Before fluorescence microscopic observation, cells were stained with a nucleic acid binding dye, acridine orange (0.005%, *w*/*v*). Fluorescence from acridine orange and chlorophylls were detected by 460 nm excitation and 395 to 440 nm excitation, respectively, with a long-pass emission filter. 

### 2.6. Scanning Electron Microscopy (SEM) Imaging

Cell aggregates collected from culture solutions were fixed in 4% paraformaldehyde, 4% glutaraldehyde in phosphate buffer (pH 7.4) at 4 °C for 1 h and then 2% glutaraldehyde in phosphate buffer for one night and subsequently postfixed in 2% osmium tetraoxide in phosphate buffer at 4 °C for 2 h. The specimens were dehydrated in a graded ethanol and dried by CO_2_ critical point dry. Dried specimens were coated with an osmium plasma ion coater (OPC-80; Nippon Laser & Electronics, Tokyo, Japan). The observation was made using JSM-7500F at 5 kV (JEOL) at the Hanaichi UltraStructure Research Institute (Okazaki, Japan).

### 2.7. DNA Extraction and Quantitative PCR (qPCR) 

Bacterial cells were collected from co-culture solutions, and total DNA was extracted from planktonic cells and cell aggregates using the DNeasy UltraClean microbial kit (Qiagen) and DNeasy power biofilm kit (Qiagen), respectively, according to the manufacturer’s instructions. For cell aggregates, cells were beaten with a FastPrep 24 instrument (Funakoshi, Tokyo, Japan) for 60 s at 4 m/s before DNA extraction. PCR primer sets targeting the *rpoB* gene were designed to distinguish *Thermosynechococcus* sp. NK55a (5′-CCTCCTATTTACATGACGGCT-3′ and 5′-AGATAATCTGCACTGGCGAA-3′) and *C. aggregans* NBF (5′-GTCAGATTCTCCGTGAGGACATC-3′ and 5′-GGTTATGTTCATCGAGCGGTGCA-3′). The StepOne Real-Time PCR system (Applied Biosystems, Foster City, CA, USA) was used with FastStart Universal SYBR Green Master (Roche, Mannheim, Germany). The reaction mixture was composed of 10 μL of FastStart SYBR Green Master mix, 0.12 μL of 50 μM primers, 8.76 μL of water, and 1 μL of DNA solution. Real-time PCR was performed using the following protocol: first denaturation, 95 °C for 10 min; denaturation and amplification, 95 °C for 15 s and 60 °C for 60 s, respectively (40 cycles). For standard curves, genomic DNA was used after the concentrations were spectrophotometrically quantified with BioSpec-nano (Shimadzu). 

### 2.8. Total Protein Quantification after Size Fractionation

At the appropriate incubation time, the co-culture solution was filtered through nylon mesh filters (pore size, 20 μm), and the filters were washed with BG11 medium. Cells in the flow-through fraction, collected by centrifugation at 7600× *g* for 5 min, and cells on the filter were suspended in 0.2 mol/L of NaOH and 2% SDS solution containing zirconia silica microbeads (∅1 mm and ∅0.2 mm). The cell suspension was beaten with a FastPrep 24 instrument (Funakoshi) for 60 s at 4 m/s, incubated at 95 °C for 15 min, and centrifuged to collect supernatant. Protein concentration in the supernatant was determined with a DC protein assay kit (Bio-Rad, Hercules, CA, USA) and a spectrophotometer (Infinite 200PRO; Tecan, Seestrasse, Switzerland) using bovine serum albumin as a standard.

### 2.9. Statistical Analysis

A two-sample statistical test (Student’s *t*-test) was used to compare the amount of total cellular proteins before and after incubation. The significant difference was considered when the *p* value was <0.05.

## 3. Results

### 3.1. Cell Aggregate Formation in the Co-Culture of Thermosynechococcus sp. NK55a and C. aggregans NBF

*Thermosynechococcus* sp. NK55a and *C. aggregans* NBF were co-inoculated into BG11 medium in glass vials and incubated at 50 °C in incandescent light under gently rolling conditions. Periodic photographs of the glass vial and micrographs of the culture solution were taken (Figure 1a). At the beginning of the cultivation, cells in the culture were dispersed well. After 4 h cultivation, small and low dense cell aggregates (20–100 µm) appeared, and the number and size of cell aggregates increased. Finally, densely packed cell aggregates developed at 100 to 200 µm in diameter, and the color of the culture solution turned transparent. Cell aggregates formed after 12 h were hardly dispersed by vigorous vortex mixing. After size fractionation with a 20 μm pore size filter, the aggregation index was determined (Figure 1b). At 0 h incubation, the aggregation index was ~20%, resulting from capturing a part of long filamentous cells of *C. aggregans* on the filter. The index gradually increased during incubation and reached ~80% after 12 h incubation. No further increase in the aggregation index was observed after 12 h (80.8%, 16 h). Total cellular proteins did not significantly increase with incubation (3.02 ± 0.90 mg mL^−1^ at 0 h and 3.09 ± 0.53 mg mL^−1^ at 12 h; *t* = 0.92, Student’s *t*-test), suggesting no marked cellular growth during the incubation.

qPCR was performed to estimate the population ratio of the two species in cell aggregates. Total DNA was extracted from cell aggregates formed by 12 h incubation. The copy number of the *rpoB* gene was quantified by qPCR to calculate the genome copies of each species; (4.70 ± 0.74) × 10^8^ and (8.33 ± 1.25) × 10^8^ genome copies in cell aggregates per one vial were detected for *Thermosynechococcus* sp. NK55a and *C. aggregans* NBF, respectively. As observed in many cyanobacteria [40], polyploidy was reported for *Thermosynechococcus* sp. [41]. The genome copy number was estimated to be four to five copies per cell for the axenic culture of *Thermosynechococcus* sp. NK55a by qPCR and microscopic cell counting.

### 3.2. Morphology of Co-Aggregates

Cell aggregates were subjected to fluorescence microscopic analysis after staining with acridine orange (Figure 2a-1,a-2). Unicellular rod cells are reddish, indicating chlorophyll *a*-containing cyanobacteria. Filamentous cells were detected with a greenish color by acridine orange. The microscopic image indicated that unicellular cyanobacterial cells and filamentous cells formed cell aggregate in a mosaic manner. Kawano et al. reported that extracellular cellulose production in *Thermosynechococcus* sp. RKN was induced under low-temperature illuminated conditions to form cell aggregates, which were dispersed by cellulase treatment [29]. Co-aggregates formed by *Thermosynechococcus* sp. NK55a and *C. aggregans* NBF were not dispersed by cellulase treatment (Appendix A).

Cell aggregates formed after 12 h incubation were subjected to SEM analysis. Cyanobacterial cells were detected for bridging to filamentous cells (Figure 2b-1). Focusing on the interface between unicellular cells and filamentous cells, thin fibers (20–30 nm in diameter) were observed (Figure 2b-2). The fibers seemed to be type IV pili, which were identified at both poles of *Thermosynechococcus* cell [22]. 

### 3.3. Co-Cultivation of the *∆*pilB Mutant of Thermosynechococcus sp. and C. aggregans NBF

SEM analysis of cell aggregates (Figure 2b-2) expected that pili from *Thermosynechococcus* sp. contributed to cell aggregate formation with *C. aggregans*. The pili-less mutant of *Thermosynechococcus* sp. NK55a was constructed to evaluate the effects on cell aggregation. The *pilB* gene encodes cytoplasmic ATPase, essential to assemble pilin monomers [42,43]. It was reported that disruption of the *pilB* gene (tll0122) in *Thermosynechococcus* sp. resulted in lacking pilus filaments and motility [22]. TEM analysis of the ∆*pilB* mutant strain of *Thermosynechococcus* sp. NK55a confirmed the lack of fibers on the cell surface, although cells of the wild-type strain possessed a few fibers near the cell pole (Figure 3a). Co-culture of the ∆*pilB* mutant of *Thermosynechococcus* sp. NK55a and *C. aggregans* NBF was conducted as carried out for the wild-type strain in Figure 1. The aggregation index after 12 h incubation for the mutant strain was ~30% (Figure 3b), and no marked cell aggregate formation was visually observed (Figure 3c). Micro cell aggregates were microscopically detected, but cell aggregates were easily dispersed by hand shaking the glass vials. 

### 3.4. Effects of Illumination Wavelength on Cell Aggregate Formation

Co-cultivation of *Thermosynechococcus* sp. NK55a (wild-type) and *C. aggregans* NBF in the dark resulted in no marked cell aggregate formation (Figure 4). The effects of illumination wavelength on cell aggregate formation by the co-culture were examined. Cells of these phototrophs, *Thermosynechococcus* sp. and *C. aggregans*, are known to show the absorption peaks at ~630 nm and ~740 nm, respectively [24]. After 12 h incubation under illumination at 630 nm and 740 nm LED arrays, tight cell aggregates 100 to 200 µm in diameter were formed, and the color of the culture solution was transparent (Figure 4). Cell aggregate formation was similarly observed after incubation at 740 nm illumination (Figure 4). Under only 630 nm illumination, cell aggregates were formed (50 to 200 µm in diameter) (Figure 4), but cell aggregates were loosely packed and easily dispersed by hand shaking the vials. 

## 4. Discussion

In this study, the co-culture of a piliated unicellular cyanobacterium *Thermosynechococcus* sp. with a filamentous gliding bacterium *C. aggregans* formed tightly packed cell aggregates (Figure 1). Tight cell aggregate formation was observed under 740 nm LED arrays (Figure 4), indicating that cyanobacterial active photosynthesis was not necessary, but photosynthetic activity of *C. aggregans* was required. It is likely that ATP produced through cyclic photophosphorylation was mainly utilized for the gliding motility of *C. aggregans* [33] because electron and carbon sources for *C. aggregans* were limited in the culture. Based on the results, this study proposed cell aggregate formation steps as follows (Figure 5): piliated cyanobacterial cells attach to *Chloroflexus* filamentous cells. *Chloroflexus* filaments attached to cyanobacterial cells gather together through gliding motility, and motility draws filamentous cells close to each other. The proximity enables cyanobacterial cells to bridge filamentous cells. The gliding motility of *Chloroflexus* filaments and cross-linking via cyanobacterial cells results in tightly and densely packed cell aggregate formation.

Co-culture of the ∆*pilB* mutant of *Thermosynechococcus* sp. with *Chloroflexus* did not form tight cell aggregates (Figure 3b,c). The *pilB* gene encodes a key unit required for pili assembly [44,45]. As shown by TEM images (Figure 3a), pili and pili-like structures were not detected at the cell surface of the mutant strain. In cyanobacteria species, pili which is clearly detectable by TEM is called as “thick pili” (5 to 8 nm width and ≥1 µm length) [46,47]. Thick pili are the machinery for twitching motility in unicellular cyanobacteria [44,46] and adhere to the solid surface. In *Thermosynechococcus*, a few pili were detected near the cell poles in strain NK55a (Figure 3a), as reported for another strain of *Thermosynechococcus* [22]. To the authors’ knowledge, no report has detected cell-to-cell adhesion via thick pili. Pili detected on cyanobacterial cells by SEM analysis in the co-culture (Figure 2b-2) could be thick pili and likely worked to connect filamentous cells. The number of pili detected by the SEM analysis was limited. This may be due to the detachment of a part of pili during cell treatment for SEM analysis, as reported previously [48]. 

In several cyanobacterial species, the existence of so-called “thin pili” was suggested [22,49,50]. The structure and function of thin pili remain unclear in most cyanobacterial species [3]. In *Synechocystis* sp., the disruption of a gene encoding thin pili structural protein reduced the floc-forming ability [51,52]. Genome analyses of *Thermosynechococcus* suggested that it harbors multiple copies of the gene encoding the structural protein of pili. ∆*pilB* mutant cells of *Thermosynechococcus* sp. could also lose other types of pili than thick pili. However, it was hard to identify thin pili by SEM observation and distinguish genes for thin pili from the genes for thick pili. 

PilB for type IV pilus assembly shows homology with a secretion ATPase required for a type II secretion system [53,54]. In *Synechococcus elongatus* PCC7942 and *Synechocystis* PCC 6803, it was suggested that PilB also works for secretion of extracellular substances [3]. However, the deficiency of PilB did not markedly change the cell morphology and adhesive properties of *Thermosynechococcus* sp. under axenic conditions. 

Cell aggregate formation by extracellular polymeric substances has not been reported for *Chloroflexus* [30,32,55,56,57]. Non-adhesive cell aggregate formation of *C. aggregans* via gliding motility was described in 1995 [32], but the gliding machinery of *C. aggregans* has not been identified yet [58]. Adhesive cell aggregate formation with *Thermosynechococcus* detected in this study probably required the gliding motility of *C. aggregans* as described above. This idea was supported by the observation of no cell aggregate formation in the co-culture of *Thermosynechococcus* with a slow gliding strain, *Chloroflexus aurantiacus* J-10-fl (Appendix A), whose gliding rate was ~1/100 slower than that of *C. aggregans* [30,32,33]. The gliding motility of *C. aggregans* could be affected by interspecies interactions [33,35]. Hanada et al. suggested that cAMP provided by cyanobacteria possibly promotes the gliding motility of *C. aggregans* [33]. 

This study found that *Thermosynechococcus* unicellular cyanobacteria formed densely packed cell aggregates by the co-cultivation with filamentous bacteria in *Chloroflexus*. This study indicated that *Chloroflexus* active gliding movement entangled piliated cyanobacterial cells in the filament matrix, increased cell density, and promoted cross-linking for cell aggregate formation. This resulted in the stimulation of interspecies interactions, such as the supply of fixed carbons by cyanobacteria to the heterotrophic growth of *Chloroflexus* and the removal of excess O_2_ by *Chloroflexus* for cyanobacterial growth. Immobilization of densely packed microbial cells will facilitate the application in bioproduction [59,60,61].

## Figures and Tables

**Figure 1 microorganisms-12-01904-f001:**
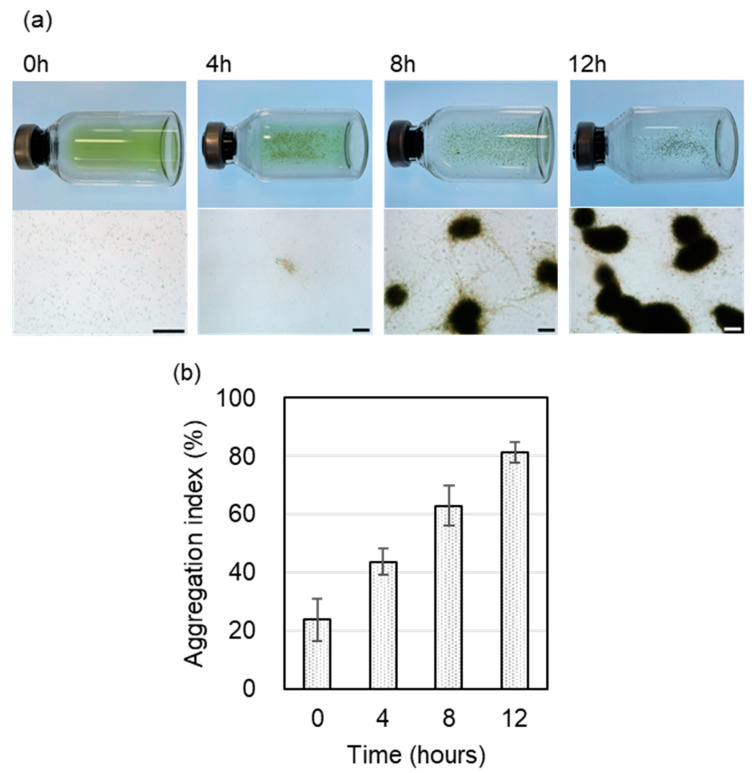
Co-cultivation of *Thermosynechococcus* sp. NK55a and *C. aggregans* NBF under incandescent light. *Thermosynechococcus* sp. NK55a and *C. aggregans* NBF were co-inoculated into 5 mL of BG11 medium in a 25 mL glass vial and cultivated at 50 °C in incandescent light. (**a**) Photographs of the glass vial (top) and bright-field micrographs of the culture solution (bottom; bars, 100 μm) after 0 h, 4 h, 8 h, and 12 h incubation. (**b**) The cellular protein amount was determined for the filtrate and residue after size fractionation by filtration (pore size, 20 μm), and the percentage of the protein amount of residue on the filter in the total amount (filtrate and residue) was calculated as the aggregation index. Each bar indicates the average of three independent cultivations with three replicates. Error bars indicate standard deviations.

**Figure 2 microorganisms-12-01904-f002:**
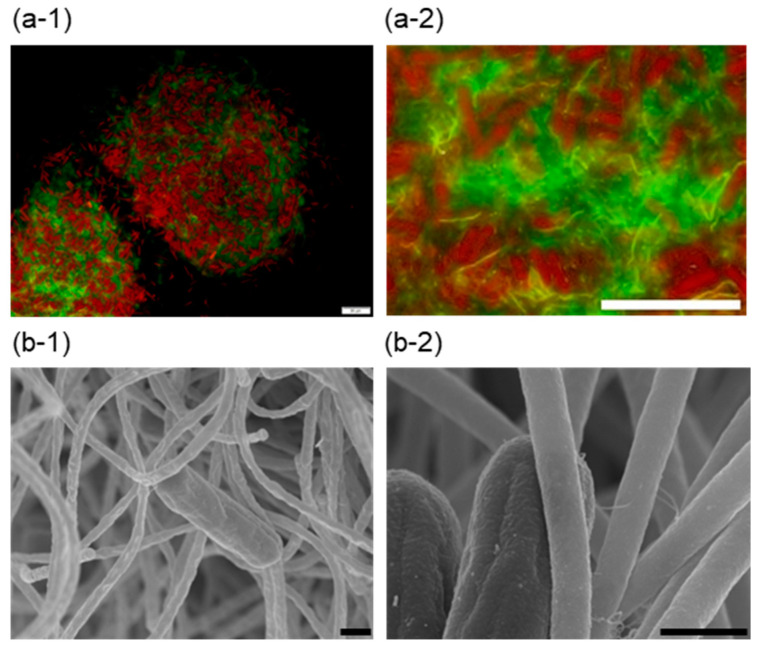
Microscopic observation of cell aggregates in the co-culture. (**a-1**,**a-2**) Fluorescence micrographs of cell aggregates after 16 h incubation with acridine orange staining. Reddish and greenish colors were detected by 460 nm excitation and 395 to 440 nm excitation, respectively, with a long-pass emission filter. Bars, 20 µm. (**b-1**,**b-2**) SEM images of cell aggregates after 12 h incubation. Bars, 1 µm.

**Figure 3 microorganisms-12-01904-f003:**
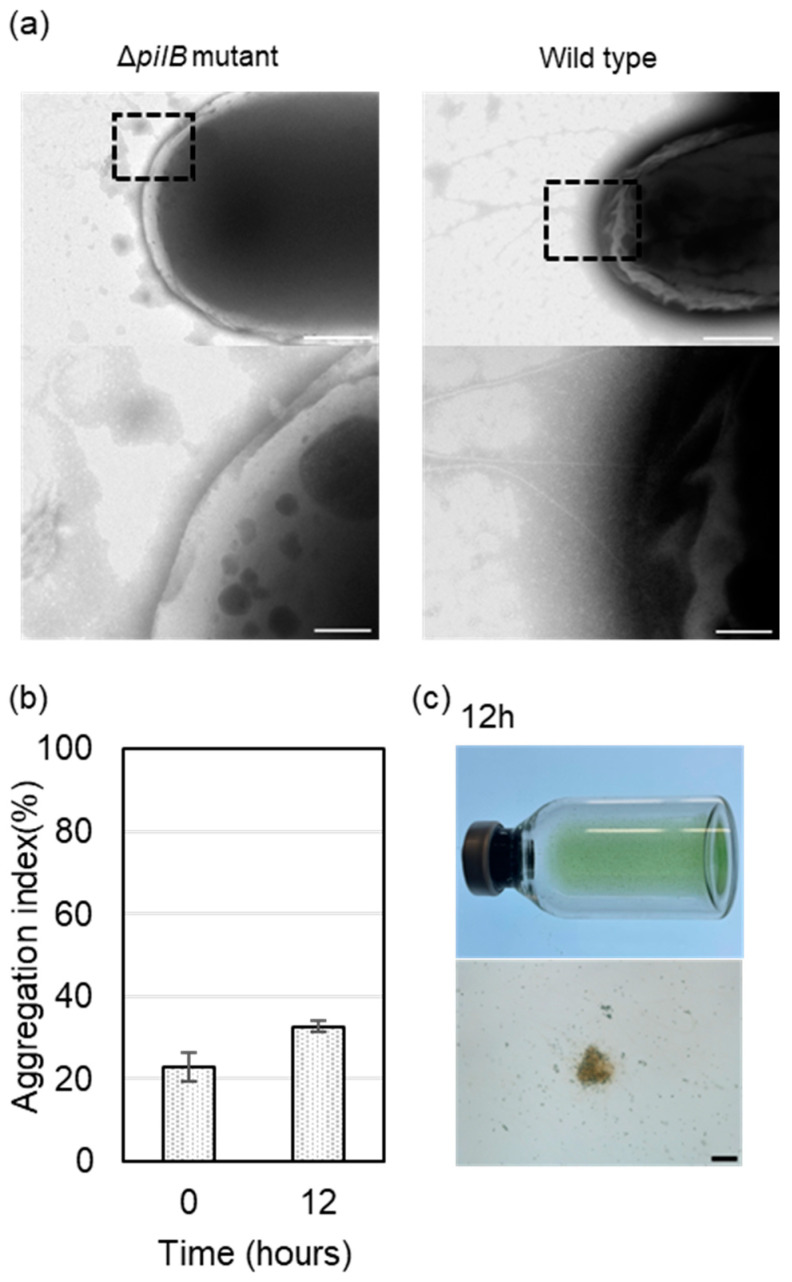
Analyses of the Δ*pilB* mutant strain of *Thermosynechococcus* sp. NK55a. (**a**) TEM images of cells of the Δ*pilB* mutant strain of *Thermosynechococcus* sp. NK55a (Δ*pilB* mutant) and the wild-type strain (Wild type). Top, bars, 500 nm; bottom, close-up view of the area surrounded by a dotted square in the top images, bars, 100 nm. (**b**) The Δ*pilB* mutant strain and *C. aggregans* NBF were co-cultivated, and aggregation indexes just before cultivation and after 12 h cultivation were determined as carried out in Figure 1. Each bar indicates the average of three independent cultivations with three replicates. Error bars indicate standard deviations. (**c**) Photographs of the glass vial (top) and bright-field micrographs of the culture solution (bottom: bars, 100 μm) after 12 h co-cultivation of the Δ*pilB* mutant strain and *C. aggregans* NBF.

**Figure 4 microorganisms-12-01904-f004:**
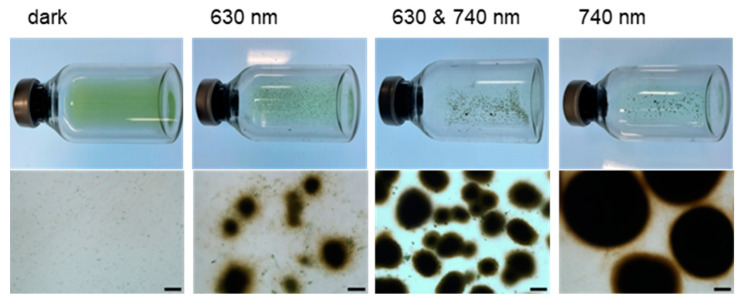
Co-cultivation of *Thermosynechococcus* sp. NK55a and *C. aggregans* NBF in the dark and LED arrays. *Thermosynechococcus* sp. NK55a and *C. aggregans* NBF were co-cultivated for 12 h as in Figure 1 but under different illumination conditions: dark, 630 nm LED array, 630 nm and 740 nm LED arrays, and 740 nm LED array. Top, photographs of the glass vials; bottom, bright-field micrographs of the culture solution. Bars, 100 μm.

**Figure 5 microorganisms-12-01904-f005:**
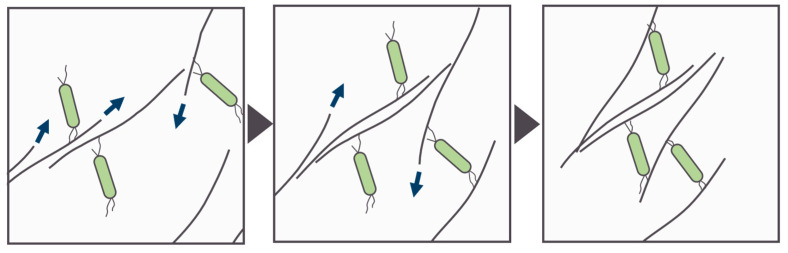
Schematic drawing of cell aggregate formation steps (from left to right) by *C. aggregans* (thin filaments) with *Thermosynechococcus* sp. (piliated rod-shaped cell). *Thermosynechococcus* cells attach to *C. aggregans* filaments via pili. The gliding motility of *C. aggregans* hauls in its filamentous cells. *C. aggregans* filaments are cross-linked by *Thermosynechococcus* cells to form firmly packed cell aggregates. Arrows indicate the direction of the gliding motility of *C. aggregans* filaments.

**Table 1 microorganisms-12-01904-t001:** PCR primers for construction of pUC19-ΔpilB_CmR.

Product (Size)	Name	Sequence (5′–3′)
pUC19 vector backbone (2684 bp)	pUC19_1R1	GGCTTCTTCCCTAGAGTGCAAGCTTGGCGTAA
	pUC19_8F1	CTGTTTTGGCTGACTACCGAGCTCGAATTCAC
Chloramphenicol-resistant cassette of pHSG398 (1093 bp)	CmR_4F1	TCGATCCCGAACTACGGAAGATCACTTCGCAG
	CmR_5R1	AAGTCTACACGTCCTCACATTAATTGCGTTGC
Upstream region of *pilB* (2531 bp)	up NKpilB_2F1	CCAAGCTTGCACTCTAGGGAAGAAGCCAGCG
	up NKpilB_3R1	GTGATCTTCCGTAGTTCGGGATCGAGGCAATC
Downstream region of *pilB* (2607 bp)	dw NKpilB_6F1	GCAACGCAATTAATGTGAGGACGTGTAGACTTC
	dw NKpilB_7R1	TTCGAGCTCGGTAGTCAGCCAAAACAGCGATCC

The underlined sequence is specific to the target gene.

## Data Availability

The original contributions presented in the study are included in the article/Appendix A, further inquiries can be directed to the corresponding author.

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
