# Peer review of "Coaggregation Occurs between a Piliated Unicellular Cyanobacterium, Thermosynechococcus, and a Filamentous Bacterium, Chloroflexus aggregans"

_microorganisms, 2024, doi:10.3390/microorganisms12091904_

Round 1
Reviewer 1 Report
Comments and Suggestions for Authors
In reviewing MS very interesting and new to science results about co-aggregation between unicellular cyanobacterium and filamentous bacterium are presented. I am sure that the paper will be of interest to microbiologists. The authors were able to detect cell-to-cell adhesion via thick pili, using microbiological, molecular-genetic methods, and scanning-electron microscopy. All sections of the MS are well written and clear. The data is supported by high-quality figures. The basic results of the paper are clearly presented in Figure 5. Conclusions are based on the results.
I recommending the MS for the publication after some corrections.
Major comments:
1. Materials and methods: Add information about ecological conditions of Nakabusa Hot Springs (temperatures, pH, and others parameters).
2. Add recent publications (within the last 5 years) to the reference list. You have only 14 newly published references in it.
Minor comments:
Title: Please, add information about authors of the taxa.
Line 29: Add reference on the first sentence.
Line 54 and further: Add authors for the genus and species names at first mention.
Figures 1 b, 2a, 4: Please, move the figures to the center of the page.
Author Response
[Our reply to comments from the reviewers]
In the following, we have put the original comments in bold and our replies are in normal text.
In reviewing MS very interesting and new to science results about co-aggregation between unicellular cyanobacterium and filamentous bacterium are presented. I am sure that the paper will be of interest to microbiologists. The authors were able to detect cell-to-cell adhesion via thick pili, using microbiological, molecular-genetic methods, and scanning-electron microscopy. All sections of the MS are well written and clear. The data is supported by high-quality figures. The basic results of the paper are clearly presented in Figure 5. Conclusions are based on the results. I recommending the MS for the publication after some corrections.
Major comments:
- Materials and methods: Add information about ecological conditions of Nakabusa Hot Springs (temperatures, pH, and others parameters).
We have added in Introduction.
- Add recent publications (within the last 5 years) to the reference list. You have only 14 newly published references in it.
We added the following recent publications in Introduction and Discussion.
Çoruh, O.; Frank, A.; Tanaka, H.; Kawamoto, A.; El-Mohsnawy, E.; Kato, T.; Namba, K.; Gerle, C.; Nowaczyk, M. M.; Kurisu, G. Cryo-EM structure of a functional monomeric Photosystem I from Thermosynechococcus elongatus reveals red chlorophyll cluster. Commun Biol 2021, 4, 304. DOI: 10.1038/s42003-021-01808-9.
Huang, G.; Xiao, Y.; Pi, X.; Zhao, L.; Zhu, Q.; Wang, W.; Kuang, T.; Han, G.; Sui, S. F.; Shen, J. R. Structural insights into a dimeric Psb27-photosystem II complex from a cyanobacterium Thermosynechococcus vulcanus. Proc Natl Acad Sci U S A. 2021, 118, e2018053118. DOI: 10.1073/pnas.2018053118.
Kawakami, K.; Nagao, R.; Tahara, Y. O.; Hamaguchi, T.; Suzuki, T.; Dohmae, N.; Kosumi, D.; Shen, J. R.; Miyata, M.; Yonekura, K.; Kamiya, N. Structural implications for a phycobilisome complex from the thermophilic cyanobacterium Thermosynechococcus vulcanus. Biochim Biophys Acta Bioenerg 2021, 1862, 148458. DOI: 10.1016/j.bbabio.2021.148458.
Haruta, S.; Kakuhama, H.; Fukushima, S.; Morohoshi, S. Motility assay of Chloroflexus. In: Bacterial and Archaeal Motility (Springer Protocols, Methods in Molecular Biology). Minamino, T.; Miyata, M.; Namba, K. Eds. Humana New York, NY, 2023, pp. 383-390
Kamagata, Y. Recent biofilm studies open a new door in microbial ecology. Microbes Environ 2020, 35, ME3501rh. DOI: 10.1264/jsme2.ME3501rh
Mittermeier, F.; Bäumler, M.; Arulrajah, P.; García Lima, J. J.; Hauke, S.; Stock, A.; Weuster-Botz, D. Artificial microbial consortia for bioproduction processes. Eng Life Sci 2022, 23, e2100152. DOI: 10.1002/elsc.202100152.
Li, Z.; Wang, X.; Wang, J.; Yuan, X.; Jiang, X.; Wang, Y.; Zhong, C.; Xu, D.; Gu, T.; Wang, F. Bacterial biofilms as platforms engineered for diverse applications. Biotechnol Adv 2022, 57, 107932. DOI: 10.1016/j.biotechadv.2022.107932.
Minor comments:
Title: Please, add information about authors of the taxa.
We would like to follow International Code of Nomenclature of Prokaryotes (ICNP) to describe the taxa of bacteria used in this study, since many previous articles applied ICNP to Thermosynechococcus and Chloroflexus. But, to make authors who describe the taxa clear, we added references in the text (please see below).
Line 29: Add reference on the first sentence.
We have added.
Line 54 and further: Add authors for the genus and species names at first mention.
As mentioned above, we would like to follow International Code of Nomenclature of Prokaryotes (ICNP) to describe the taxa of bacteria used in this study, since many previous articles applied ICNP to Thermosynechococcus and Chloroflexus. But, to make authors who describe the taxa clear, we added references.
In slightly alkaline hot-spring water from 50°C to 65°C, a deeply branched cyanobacterial group, Thermosynechococcus (H.Katoh, S.Itoh, J.-R.Shen & M.Ikeuchi [REF]), is widely distributed as a dominant species worldwide [11,12].
(REF) Katoh, H.; Itoh, S.; Shen, J. R.; Ikeuchi, M. Functional analysis of psbV and a novel c-type cytochrome gene psbV2 of the thermophilic cyanobacterium Thermosynechococcus elongatus strain BP-1. Plant Cell Physiol 2001, 42, 599-607. DOI: 10.1093/pcp/pce074.
Bacteria in the genus Chloroflexus (Pierson & Castenholz [REF]) have been characterized as a fast-gliding bacteria [25]
(REF) Pierson, B. K.; Castenholz, R. W. A phototrophic gliding filamentous bacterium of hot springs, Chloroflexus aurantiacus, gen. and sp. nov. Arch Microbiol 1974, 100, 5-24. DOI: 10.1007/BF00446302.
The representative species, Chloroflexus aggregans (Hanada et al. [REF]) forms cell aggregates through active gliding motility on other filaments [26],
(REF) Hanada, S.; Hiraishi, A.; Shimada, K.; Matsuura, K. Chloroflexus aggregans sp. nov., a filamentous phototrophic bacterium which forms dense cell aggregates by active gliding movement. Int J Syst Bacteriol 1995, 45, 676-681. DOI: 10.1099/00207713-45-4-676.
Figures 1 b, 2a, 4: Please, move the figures to the center of the page.
We have done.
Reviewer 2 Report
Comments and Suggestions for Authors
Comments on microorganisms-3180569 entitled “Coaggregation occurs between a piliated unicellular cyanobacterium, Thermosynechococcus and a filamentous bacterium, Chloroflexus aggregans”
The manuscript is devoted to coaggregation between unicellular cyanobacterium, Thermosynechococcus H.Katoh, S.Itoh, J.-R.Shen & M.Ikeuchi (Acaryochloridales, Cyanobacteria) and a filamentous bacterium, Chloroflexus aggregans Hanada et al. (Chloroflexales, Chloroflexota), isolated from Nakabusa Hot Springs (Nagano, Japan). The study of co-aggregation and symbiosys of cyanobacteria with other heterotrophic bacteria is very important and interesting for the understanding of ecology and biology of these organisms. However, such works are still scarce. That is why this study is relevant.
However, I have some questions and remarks dealing with this manuscript:
1) It would be better if the authors corrected the title of the article to “Coaggregation occurs between a piliated unicellular cyanobacterium, Thermosynechococcus H.Katoh, S.Itoh, J.-R.Shen & M.Ikeuchi and a filamentous bacterium, Chloroflexus aggregans Hanada et al.”.
2) The authors of the species and genera are not listed at first mention in text; it should be corrected. For example:
Line 41: “Thermosynechococcus” should be corrected to “Thermosynechococcus H.Katoh, S.Itoh, J.-R.Shen & M.Ikeuchi”,
etc.
3) Line 53: “Castenholz (2015) suggested” should be corrected to “Castenholz in 2015 suggested” or “Castenholz suggested”.
4) Line 118: In subheading 2.4 the authors write about “Co-cultivation of Thermosynechococcus sp. and Chloroflexus sp.”. But above they provide data that a strain Chloroflexus aggregans NBF was used for the experiment (lines 72-73). I would like to clarify what species they used, Chloroflexus sp. or Chloroflexus aggregans?
5) A subheading "Statistical Analysis" should be added to “Materials and Methods” for better understanding of the text. Although the authors provide some statistical data in “Results” (for example, in the caption to Figure 1), this presentation makes the article difficult to read.
6) The color of the scale bars in the Figures 1, 3, 4 should be changed from white to black, as they are almost invisible.
Author Response
[Our reply to comments from the reviewers]
In the following, we have put the original comments in bold and our replies are in normal text.
The manuscript is devoted to coaggregation between unicellular cyanobacterium, Thermosynechococcus H.Katoh, S.Itoh, J.-R.Shen & M.Ikeuchi (Acaryochloridales, Cyanobacteria) and a filamentous bacterium, Chloroflexus aggregans Hanada et al. (Chloroflexales, Chloroflexota), isolated from Nakabusa Hot Springs (Nagano, Japan). The study of co-aggregation and symbiosys of cyanobacteria with other heterotrophic bacteria is very important and interesting for the understanding of ecology and biology of these organisms. However, such works are still scarce. That is why this study is relevant. However, I have some questions and remarks dealing with this manuscript:
1) It would be better if the authors corrected the title of the article to “Coaggregation occurs between a piliated unicellular cyanobacterium, Thermosynechococcus H.Katoh, S.Itoh, J.-R.Shen & M.Ikeuchi and a filamentous bacterium, Chloroflexus aggregans Hanada et al.”.
We would like to follow International Code of Nomenclature of Prokaryotes (ICNP) to describe the taxa of bacteria used in this study, since many previous articles applied ICNP to Thermosynechococcus and Chloroflexus. But, to make authors who describe the taxa clear, we added references in the text (please see below).
2) The authors of the species and genera are not listed at first mention in text; it should be corrected. For example:
Line 41: “Thermosynechococcus” should be corrected to “Thermosynechococcus H.Katoh, S.Itoh, J.-R.Shen & M.Ikeuchi”,
etc.
As mentioned above, we would like to follow International Code of Nomenclature of Prokaryotes (ICNP) to describe the taxa of bacteria used in this study, since many previous articles applied ICNP to Thermosynechococcus and Chloroflexus. But, to make authors who describe the taxa clear, we added references.
In slightly alkaline hot-spring water from 50°C to 65°C, a deeply branched cyanobacterial group, Thermosynechococcus (H.Katoh, S.Itoh, J.-R.Shen & M.Ikeuchi [REF]), is widely distributed as a dominant species worldwide [11,12].
(REF) Katoh, H.; Itoh, S.; Shen, J. R.; Ikeuchi, M. Functional analysis of psbV and a novel c-type cytochrome gene psbV2 of the thermophilic cyanobacterium Thermosynechococcus elongatus strain BP-1. Plant Cell Physiol 2001, 42, 599-607. DOI: 10.1093/pcp/pce074.
Bacteria in the genus Chloroflexus (Pierson & Castenholz [REF]) have been characterized as a fast-gliding bacteria [25]
(REF) Pierson, B. K.; Castenholz, R. W. A phototrophic gliding filamentous bacterium of hot springs, Chloroflexus aurantiacus, gen. and sp. nov. Arch Microbiol 1974, 100, 5-24. DOI: 10.1007/BF00446302.
The representative species, Chloroflexus aggregans (Hanada et al. [REF]) forms cell aggregates through active gliding motility on other filaments [26],
(REF) Hanada, S.; Hiraishi, A.; Shimada, K.; Matsuura, K. Chloroflexus aggregans sp. nov., a filamentous phototrophic bacterium which forms dense cell aggregates by active gliding movement. Int J Syst Bacteriol 1995, 45, 676-681. DOI: 10.1099/00207713-45-4-676.
3) Line 53: “Castenholz (2015) suggested” should be corrected to “Castenholz in 2015 suggested” or “Castenholz suggested”.
We have corrected.
4) Line 118: In subheading 2.4 the authors write about “Co-cultivation of Thermosynechococcus sp. and Chloroflexus sp.”. But above they provide data that a strain Chloroflexus aggregans NBF was used for the experiment (lines 72-73). I would like to clarify what species they used, Chloroflexus sp. or Chloroflexus aggregans?
We have modified to make the species clear.
5) A subheading "Statistical Analysis" should be added to “Materials and Methods” for better understanding of the text. Although the authors provide some statistical data in “Results” (for example, in the caption to Figure 1), this presentation makes the article difficult to read.
We have added as follows.
2.9. Statistical analysis
A two-sample statistical test (Student’s t-test) was used to compare the amount of total cellular proteins before and after incubation. The significant difference was considered when the P value was <0.05.
6) The color of the scale bars in the Figures 1, 3, 4 should be changed from white to black, as they are almost invisible.
We have modified to show the bars clearly.
Reviewer 3 Report
Comments and Suggestions for Authors
Dear Authors,
It was a pleasure to read your manuscript. The work is well-designed, well-written and the results are interesting. However, I suggest the following improvements to enhance the manuscript:
Line 24: For the term "filamentous," consider using different keywords than those in the title. Also, please check the journal's guidelines for keyword length.
Lines 63 to 65: The manuscript would benefit from explicitly postulating the hypothesis of the work.
Figure 2: Since pigments can interfere with fluorescence microscopy using AO, do the authors have microscopy images using the same excitation wavelength but without acridine orange staining? Additionally, are there higher magnification acridine orange microscopy images that allow for clearer visualization of unicellular and filamentous cells?
Figure 3: Use the same magnification for both the mutant and wild-type strains in Figure 3A. This will facilitate a better comparison of the differences.
Discussion: Provide more references to support and compare your data, offering deeper insights. The literature suggests that PilB may regulate exopolysaccharide production in some bacterial species. How might this account for your results? Discussing the interaction between pili and EPS in the context of biofilm or cell surface interactions would enrich the paper. Only include references if they genuinely enhance the manuscript; otherwise, seek additional references to discuss this possibility. Consider the following references for this discussion:
https://doi.org/10.1007/978-3-030-75289-7_3
https://www.nature.com/articles/s41598-017-07594-x
https://journals.asm.org/doi/10.1128/jb.06052-11
What practical aspects arise from your results? Do they have any applications in bioprocesses? A brief discussion on the practical considerations of your findings would benefit the article. Additionally, immobilized cultivation of microorganisms or consortia is a trending topic; consider discussing the significance of your results in this context.
Conclusion: The manuscript would be improved by including a clear conclusion of the key findings of the work.
Author Response
[Our reply to comments from the reviewers]
In the following, we have put the original comments in bold and our replies are in normal text.
It was a pleasure to read your manuscript. The work is well-designed, well-written and the results are interesting. However, I suggest the following improvements to enhance the manuscript:
Line 24: For the term "filamentous," consider using different keywords than those in the title. Also, please check the journal's guidelines for keyword length.
We would like to keep “filamentous anoxygenic photosynthetic bacteria” as one of the keywords, since this is a term to describe a group of photosynthetic bacteria, which is commonly explained in textbooks such as Brock Biology of Microorganisms (Madigan et al. Peason). We think that the keywords, “filamentous anoxygenic photosynthetic bacteria” will successfully lead the readers to understand what organism was used in this article. Instructions for Authors do not mention the length of keywords.
Lines 63 to 65: The manuscript would benefit from explicitly postulating the hypothesis of the work.
We have added as follows.
It was hypothesized that C. aggregans helps form cell aggregates of Thermosynechococcus through cell-to-cell interactions.
Figure 2: Since pigments can interfere with fluorescence microscopy using AO, do the authors have microscopy images using the same excitation wavelength but without acridine orange staining? Additionally, are there higher magnification acridine orange microscopy images that allow for clearer visualization of unicellular and filamentous cells?
Unfortunately, we do not have microscopy images without acridine orange staining. We could not clearly observe filamentous cells (Chloroflexus), when we did not use acridine orange.
To show clear images, we provided the same image with higher resolution and additionally provided a close-up view in the revised manuscript.
Figure 3: Use the same magnification for both the mutant and wild-type strains in Figure 3A. This will facilitate a better comparison of the differences.
We have applied the same magnification as the reviewer suggested.
Discussion: Provide more references to support and compare your data, offering deeper insights. The literature suggests that PilB may regulate exopolysaccharide production in some bacterial species. How might this account for your results? Discussing the interaction between pili and EPS in the context of biofilm or cell surface interactions would enrich the paper. Only include references if they genuinely enhance the manuscript; otherwise, seek additional references to discuss this possibility. Consider the following references for this discussion:
https://doi.org/10.1007/978-3-030-75289-7_3
https://www.nature.com/articles/s41598-017-07594-x
https://journals.asm.org/doi/10.1128/jb.06052-11
As the reviewer pointed out, we have added the following paragraph to Discussion by referring some literatures.
PilB for type IV pilus assembly shows homology with a secretion ATPase required for type II secretion system (Peabody et al. 2003; Black et al. 2017). In Synechococcus elongatus PCC7942 and Synechocystis PCC 6803, it was suggested that PilB also works for secretion of extracellular substances (Schuergers & Wilde. 2015). However, the deficiency of PilB did not markedly change the cell morphology and adhesive properties of Thermosynechococcus sp. under axenic conditions. It was also unlikely that the production of interspecies signaling molecules via PilB stimulated the cell aggregation of C. aggregans, since the cell-free culture supernatant of Thermosynechococcus sp. NK55a did not markedly promote the gliding motility of C. aggregans (data not shown).
(REFs)
Peabody, C. R.; Chung, Y. J.; Yen M. R.; Vidal-Ingigliardi, D.; Pugsley, A. P.; Saier, M. H. Type II protein secretion and its relationship to bacterial type IV pili and archaeal flagella. Microbiology 2003, 149, 3051-3072. DOI: 10.1099/mic.0.26364-0.
Black, W. P.; Wang, L.; Jing, X.; Saldaña, R. C.; Li, F.; Scharf, B. E.; Schubot, F. D.; Yang, Z. The type IV pilus assembly ATPase PilB functions as a signaling protein to regulate exopolysaccharide production in Myxococcus xanthus. Sci Rep 2017, 7, 7263. DOI: 10.1038/s41598-017-07594-x.
Schuergers, N.; Wilde, A. Appendages of the cyanobacterial cell. Life (Basel) 2015, 5, 700-715. DOI: 10.3390/life5010700.
What practical aspects arise from your results? Do they have any applications in bioprocesses? A brief discussion on the practical considerations of your findings would benefit the article. Additionally, immobilized cultivation of microorganisms or consortia is a trending topic; consider discussing the significance of your results in this context.
We have added the following sentence in Discussion by referring some literatures.
Immobilization of densely packed microbial cells will facilitate the application in bioproduction (Kamagata 2020; Mittermeier et al. 2022; Li et al. 2022).
(REFs)
Kamagata, Y. Recent biofilm studies open a new door in microbial ecology. Microbes Environ 2020, 35, ME3501rh. DOI: 10.1264/jsme2.ME3501rh
Mittermeier, F.; Bäumler, M.; Arulrajah, P.; García Lima, J. J.; Hauke, S.; Stock, A.; Weuster-Botz, D. Artificial microbial consortia for bioproduction processes. Eng Life Sci 2022, 23, e2100152. DOI: 10.1002/elsc.202100152.
Li, Z.; Wang, X.; Wang, J.; Yuan, X.; Jiang, X.; Wang, Y.; Zhong, C.; Xu, D.; Gu, T.; Wang, F. Bacterial biofilms as platforms engineered for diverse applications. Biotechnol Adv 2022, 57, 107932. DOI: 10.1016/j.biotechadv.2022.107932.
Conclusion: The manuscript would be improved by including a clear conclusion of the key findings of the work.
We have added the following sentence in the last paragraph of Discussion.
This study found that Thermosynechococcus unicellular cyanobacteria formed densely packed cell aggregates by the co-cultivation with filamentous bacteria in Chloroflexus.
Round 2
Reviewer 3 Report
Comments and Suggestions for Authors
Dear Authors, the article has been improved and may now be considered for publication.